# Potential of mesenchymal- and cardiac progenitor cells for therapeutic targeting of B-cells and antibody responses in end-stage heart failure

Patricia van den Hoogen[1], Saskia C. A. de Jager[1]*, Emma A. Mol[1,2], Arjan S. Schoneveld[3], Manon M. H. Huibers[4,5], Aryan Vink[4], Pieter A. Doevendans[6,7,8], Jon D. Laman[9], Joost P. G. Sluijter[1]

1 Laboratory of Experimental Cardiology, UMC Utrecht Regenerative Medicine Center, University Medical Center Utrecht, Utrecht, the Netherlands, 2 Laboratory of Cardiovascular Cell Biology, Department of Cell and Chemical Biology, Leiden University Medical Center, Leiden, the Netherlands, 3 Laboratory of Clinical Chemistry & Haematology, ARCADIA, University Medical Center Utrecht, Utrecht, the Netherlands, 4 Department of Pathology, University Medical Center Utrecht, Utrecht, the Netherlands, 5 Department of Genetics, University Medical Center Utrecht, Utrecht, the Netherlands, 6 Department of Cardiology, University Medical Center Utrecht, Utrecht, the Netherlands, 7 Netherlands Heart Institute, Utrecht, the Netherlands, 8 Central Military Hospital, Utrecht, the Netherlands, 9 Department of Biomedical Sciences of Cells and Systems (BSCS), University Medical Center Groningen, Groningen, the Netherlands

* s.c.a.dejager@umcutrecht.nl

**Data Availability Statement:** All relevant data are within the manuscript and its Supporting Information files.

## Abstract

Upon myocardial damage, the release of cardiac proteins induces a strong antibody-mediated immune response, which can lead to adverse cardiac remodeling and eventually heart failure (HF). Stem cell therapy using mesenchymal stromal cells (MSCs) or cardiomyocyte progenitor cells (CPCs) previously showed beneficial effects on cardiac function despite low engraftment in the heart. Paracrine mediators are likely of great importance, where, for example, MSC-derived extracellular vesicles (EVs) also show immunosuppressive properties *in vitro*. However, the limited capacity of MSCs to differentiate into cardiac cells and the sufficient scaling of MSC-derived EVs remain a challenge to clinical translation. Therefore, we investigated the immunosuppressive actions of endogenous CPCs and CPC-derived EVs on antibody production *in vitro*, using both healthy controls and end-stage HF patients. Both MSCs and CPCs strongly inhibit lymphocyte proliferation and antibody production *in vitro*. Furthermore, CPC-derived EVs significantly lowered the levels of IgG1, IgG4, and IgM, especially when administered for longer duration. In line with previous findings, plasma cells of end-stage HF patients showed high production of IgG3, which can be inhibited by MSCs *in vitro*. MSCs and CPCs inhibit *in vitro* antibody production of both healthy and end-stage HF-derived immune cells. CPC-derived paracrine factors, such as EVs, show similar effects, but do not provide the complete immunosuppressive capacity of CPCs. The strongest immunosuppressive effects were observed using MSCs, suggesting that MSCs might be the best candidates for therapeutic targeting of B-cell responses in HF.

**Funding:** This work was supported by CVON2011-12 HUSTCare grant from the Netherlands CardioVascular Research Initiative (CVON): The Dutch Heart Foundation, Dutch Federation of University Medical Centers, the Netherlands Organization for Health Research and Development, the Royal Netherlands Academy of Science, the ZonMW Translational Adult Stem Cell grant 1161002016, and by Horizon2020 ERC-2016-COG EVICARE (725229). The funders had no role in study design, data collection and analysis, decision to publish, or preparation of the manuscript.

**Competing interests:** The authors have declared that no competing interests exist.

## Introduction

Cardiovascular disease (CVD) is the most common cause of death globally with almost 18 million deaths per year [1]. A prominent CVD-subtype is ischemic heart disease (IHD), which is characterized by myocardial cell death due to prolonged ischemia [2]. After subsequent reperfusion strategies, further myocardial damage is initiated by the release of cardiac proteins, which can induce an inflammatory response [3,4]. Activated T- and B lymphocytes significantly contribute to adverse cardiac remodeling via the production of pro-inflammatory cytokines and antibodies [5–7], which can progress to severe heart failure (HF) [6,8–10].

Currently, progenitor cell therapy is gaining a lot of interest in order to regenerate the damaged heart due to their regenerative properties and the ability to differentiate into other cell types [11–13]. Mesenchymal stromal cells (MSCs) improve cardiac function by reducing scar size and increasing left ventricular ejection fraction (LVEF) with 2–4% [14,15]. However, engraftment of these cells in the heart is relatively poor, where less than 10% of the injected cells remain at the site of injection [16,17]. In addition, the few remaining cells rarely differentiate into cardiac cells [18]. In addition to their regenerative capacity, MSCs have also been shown to suppress inflammatory responses, antibody production, and fibrosis, mostly in a paracrine manner [19,20]. Important paracrine mediators are extracellular vesicles (EVs), small lipid bi-layered vesicles containing lipids, small RNAs and proteins, which are able to influence many processes including inflammation [21,22]. Multiple studies investigated the therapeutic potency of MSCs and MSC derived EVs in cardiovascular disease [13,23,24]. MSC-derived EVs were found to reduce infarct size and infiltration of immune cells into the affected myocardium after myocardial infarction (MI) in animal models [25]. These findings suggest that the use of MSC-derived EVs might be a promising strategy to restore cardiac function, however, technical difficulties in large scale production and purification of MSC-EV are still limiting the translation to the clinic [19,26]. Considering the developmental origin of endogenous cardiac-derived progenitor cells (CPCs), these cells might prove better candidates for cell therapy for cardiac repair. Endogenous CPCs were previously tested in several clinical trials where they improved cardiac function [12,27], especially when combined with MSCs [28,29]. CPCs also have immunosuppressive properties, for example by inhibiting T-cell proliferation, which is partly mediated by paracrine factors [30]. CPC-derived EVs are proposed to be of great importance as paracrine mediators of these cells [31–33]. However, the immunosuppressive capacity of CPCs or CPC-derived EVs on B cells and antibody-mediated immune responses has not been elucidated yet. Therefore, we investigated the *in vitro* inhibitory actions of CPCs and CPC-derived EVs on lymphocyte proliferation and the production of immunoglobulin subclasses, using immune cells from healthy controls and end-stage HF patients.

## Material and methods

### Culture of human-derived progenitor cells

Human bone marrow-derived mesenchymal stromal cells (MSCs) and cardiomyocyte progenitor cells (CPCs) were obtained and isolated as described before [34,35]. MSCs were cultured in MEM-alpha (Gibco, 32561–037) supplemented with 10% fetal bovine serum (Gibco, 10270–106) + 1% PenStrep (Lonza, 17-602E) + 0.2 mM L-ascorbic acid-2-phospate (Sigma A4034) + 1 ng/ml bFGF (Sigma F0291). CPCs were cultured in SP++ (25% EGM-2 (Lonza CC-3156) + 75% M199 (Gibco 31150–022) supplemented with 10% fetal bovine serum + 1% PenStrep + 1% non-essential amino acids (Lonza 13–114). Cultures were incubated at 37˚C (5% $CO_2$ and 20% $O_2$) and adherent cells were passaged when reaching 80–90% of confluency using trypsin digestion (0.25%, Lonza, CC-5012). MSCs and CPCs from fetal or adult donors were used in the co-cultures between passage 6–17.

## Isolation of CPC-derived extracellular vesicles and Western blotting

CPC-derived EVs were isolated using size-exclusion chromatography (SEC), as previously described [36]. In brief, fetal-derived CPCs were cultured until they reached a confluency of 80–90%, after which the medium was replaced with serum-free medium (M-199, Gibco 31150–022).

After 24 h, conditioned medium (CM), containing the EVs, was collected, centrifuged at 2000g for 15 min, and filtered (0.45 μm) to remove dead cells and debris. Next, CM was concentrated using 100-kDA molecular weight cut-off Amicon spin filters (Merck Milipore) and loaded onto a S400 highprep column (GE healthcare, Uppsala, Sweden) using an AKTA start (GE Healthcare) containing an UV 280 nm flow cell. Fractions containing EVs were pooled and filtered (0.45 μm) before further concentration procedures. The number of particles and mean size distribution were measured using Nanoparticle Tracking Analysis (Nanosight NS500, Malvern) as described before [36]. Protein concentration was measured using micro-BCA protein assay kit (Thermo Scientific). Vesicle markers were assessed by Western blotting (WB) as previously described [36]. EV protein fractions were loaded on pre-casted Bis-Tris protein gels (ThermoFischer, NW04125BOX) and run for 1 h at 160V. Proteins were transferred to polyvinylidene difluoride (PVDF) membranes (Millipore, IPVH00010) and stained for general EV markers [37] Alix (1:1000, Abcam, 177840), CD63 (1:1000, Abcam, 8219), CD81 (1:1000, Santa Cruz, Sc-166029), or Calnexin (1:1000, Tebu-bio, GTX101676). Proteins were detected using chemiluminescent peroxidase substrate (Sigma, CPS1120). Representative pictures S1 Fig.

## Isolation of peripheral mononuclear cells

Peripheral blood mononuclear cells (PBMCs) were isolated from fresh whole blood samples of healthy controls or end-stage HF patients, in compliance with the *declaration of Helsinki* and under approval of the Medical Ethics Committee Utrecht (METC, reference number 12/387). Written informed consent for collection and biobanking of blood samples was obtained. End-stage HF derived PBMCs were obtained from blood samples prior to heart transplantation. PBMCs were isolated using Ficoll-plaque PLUS gradient (GE life sciences, 17-1440-03) according to the manufacturers protocol. A total of $2,5x10^5$ PBMCs were added per well (48-wells plate) in RPMI-1640 medium (Lonza, BE12-702F) supplemented with 10% fetal bovine serum and 1% PenStrep.

## PBMC stimulation and co-culture with progenitor cells or purified EVs

For the stimulation of PBMCs and subsequent antibody production, a combination of IL-2 (120 IU/ml, BD Pharmingen, 554563) and PMA (0.123 ng/ml, Sigma, P8139) was used as previously described [33]. PBMCs were co-cultured for 10 days, without medium change, in 48-well plates ($2.5x10^5$ PBMC/well) with MSCs, CPCs ($5.0x10^4$ cells/well), or CPC-derived EVs (1x 10 μg = $6.3x10^{10}$ particles), immediately upon co-culture or 3x 10 μg ($6.3x10^{10}$ particles) added every 3 days of co-culture. After 10 days of co-culture, light-microscopic images were taken using an Olympus CKX41 microscope in combination with CellSense software. Non-adherent cells, containing the lymphocytes, were collected and processed for further analysis using flow cytometry (Gallios, Beckmann Coulter). Co-culture supernatant was collected, centrifuged at 500g for 10 min, aliquoted and stored at -80°C for immunoglobulin measurements.

## Lymphocyte proliferation

Flow cytometry (Gallios, Beckmann Coulter) was used to assess lymphocyte proliferation. Prior to co-culture, PBMCs were labeled with 1.5mM carboxyfluorescein succinimidyl ester

(CFSE, Sigma, 21888) as described previously [33]. In brief, PBMCs were incubated with CFSE for 10 min at 37˚C in a dark shaking bath. After 10 minutes, 5% of FBS was used to prevent further uptake. After two washing steps with PBS, PBMCs were incubated for 30 min with fluorescent antibodies, including CD3 for T cells (Brilliant Violet 510, Biolegend, 317332) and CD19 for B cells (PE/Cy5, Biolegend 302210). After washing with PBS, PBMCs were incubated for 30 min with a fixable viability dye (eFluor506, Bioscience, 65-0866-14) to exclude dead cells. Prior to culture, general cell composition per donor was assessed by measuring the percentage of CD3+ T cells and CD19+ B cells, to ensure that the cell populations were similar between the different donors at baseline. Lymphocyte proliferation was calculated by measuring CFSE intensity and the number of cells present in each division as described before [33]. Since we encountered some donor variations in the absolute number of proliferating cells in the stimulation assays, the stimulated PBMC condition was considered as maximum response and defined as 100% proliferation (ratio = 1) and used for normalization of the data per donor and per experiment. Data was analyzed using Kaluza Analysis Software (Beckman Coulter, version 1.3).

## Immunoglobulin multiplex

The levels of IgM and IgG subclasses (IgG1, IgG2, IgG3, IgG4) in the co-cultures (5x diluted) were measured using a Bio-Plex Pro™ human isotyping immunoassay 6-plex (Bio-Rad, 171A3100M) according to manufacturer's instructions and were all within the detection limit of the assay. Immunoglobulin levels in the supernatant after co-culture with MSC/CPC or CPC-derived EVs were calculated using internal standards included in the assay. Immunoglobulin levels are represented as relative production, with the stimulated PBMC condition defined as 100% antibody production (ratio = 1) and used for normalization of the data per donor and experiment.

## Statistics

Statistical analysis and data representation were performed using IBM SPSS Statistics 21 and Graphpad Prism (GraphPad Software Inc. version 8.01, San Diego CA, USA). Normal data distribution was tested using the Kolmogorov-Smirnov test. Group comparison was performed by a one-way ANOVA or Kruskal-Wallis test, corrected for multiple comparison testing. Each individual PBMC donor is considered as an independent individual experimental number (n), ranging from 2–8 donors per experiment. Data was considered significant with two-tailed p-values <0.05 and is presented as mean ± SEM.

## Results

### Progenitor cells suppress lymphocyte proliferation upon cell-cell contact

To investigate the immunosuppressive effects of progenitor cells on the proliferation of lymphocytes, a co-culture using MSCs or CPCs was performed (**Fig 1**). To represent normal lymphocyte activation by antigen-presenting cells, the total PBMC population was used. After 10 days of co-culture, large clusters of proliferating T cells were visible upon stimulation with IL-2 and PMA. These large clusters were smaller or even absent when PBMCs were cultured in the presence of MSCs or CPCs (**Fig 1A**). Flow cytometry was used to measure CFSE intensity and to assess lymphocyte proliferation (**Fig 1B and 1C**). FACS plots clearly showed active cell proliferation upon stimulation with IL-2 and PMA and suppression of proliferation when PBMCs were cultured with MSCs or CPCs. Quantification showed that both MSCs and CPCs significantly decreased proliferation of lymphocytes by 64±18.6% and 19±12.5% respectively (MSC p<0.0001, CPC p<0.05).

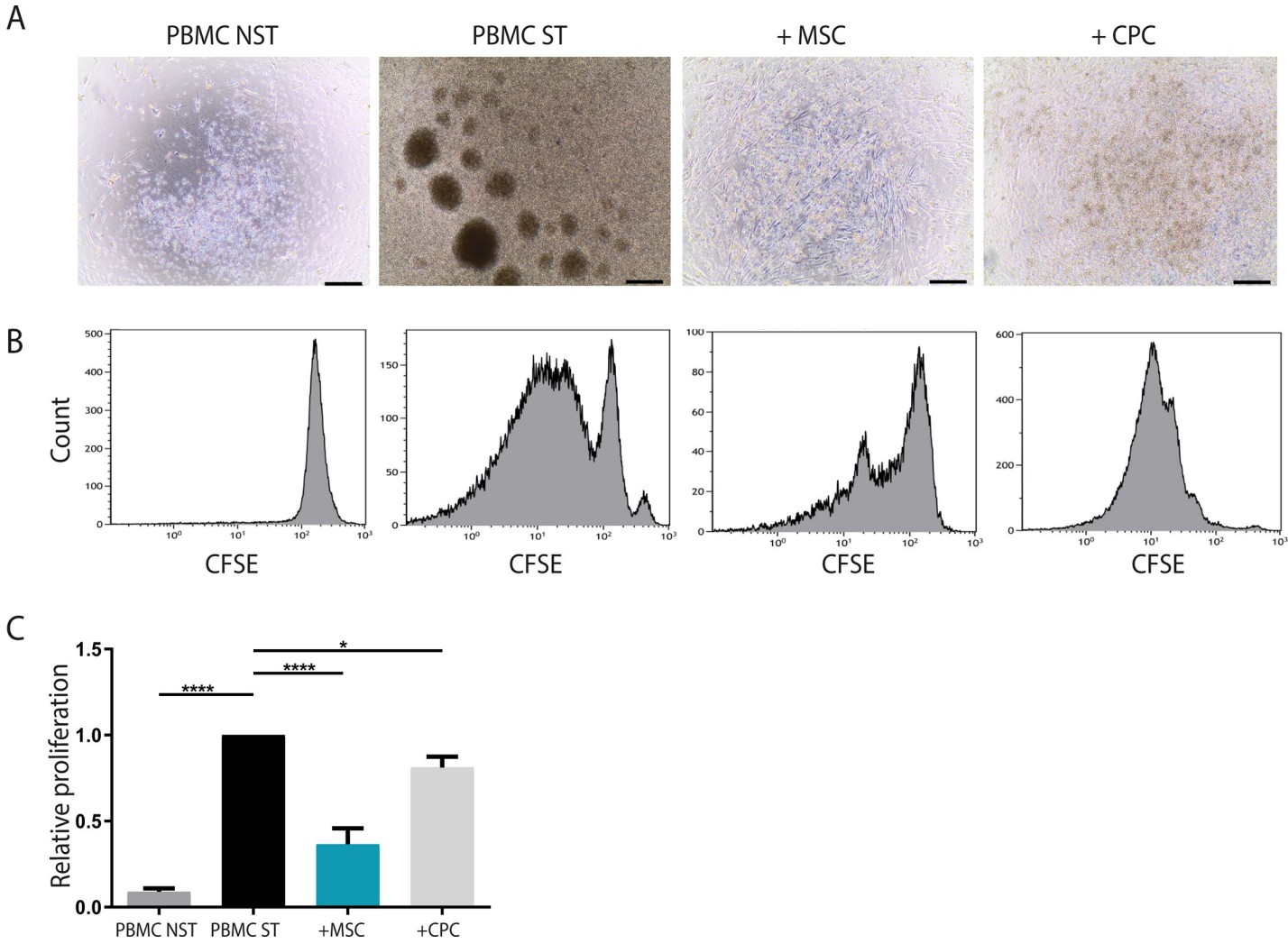

**Fig 1. Progenitor cells suppress lymphocyte proliferation.** Lymphocyte proliferation was measured after 10 days of co-culture of PBMCs with MSCs or CPCs. **A)** Representative microscopic images after 10 days of co-culture. Upon PBMC stimulation, large clusters of proliferating cells were observed. These large clusters were absent in the presence of MSCs or CPCs. **B)** PBMCs were labeled with CFSE and lymphocyte proliferation was assessed by measuring CFSE intensity using flow cytometry. FACS plots of non-stimulated lymphocytes show one peak of undivided cells, whereas upon stimulation, lymphocytes start to divide. **C)** Quantification of lymphocyte proliferation, where stimulated lymphocytes were used as normalization. Both MSCs and CPCs show a significant decrease of lymphocyte proliferation upon co-culture. Strongest effects were observed using MSCs, where proliferation was inhibited towards 36% compared to CPCs (81%). *PBMC: pheripheral blood mononuclear cells, MSC: mesenchymal stromal cell, CPC: cardiac progenitor cell. Per condition n = 4. Line bar indicates 200μm, magnification 4x. Significance was determined using one-way ANOVA, * p<0.05, **** p<0.0001.*

### Production of IgM and different IgG subclasses is suppressed by cardiac-derived progenitor cells

Next to reduced cell proliferation, MSCs are also able to inhibit several immune cell functions, such as antibody secretion [38]. To examine whether this also holds true for CPCs, we collected the supernatant after 10 days of co-culture and measured the levels of different immunoglobulin subclasses (**Fig 2A**). Since it is known that the age of the donor can affect their inhibitory potency [39,40], both fetal and adult MSCs and CPCs were included. Adult and fetal-derived MSCs significantly inhibited antibody production from stimulated PBMCs (**Fig 2B–2F**). Fetal and adult MSC significantly reduced the production of IgM (aMSC = 0.005±0.0 fMSC = 0.02±0.0; p<0.0001), IgG1 (aMSC = 0.24±0.06, fMSC = 0.28±0.06; p<0.0001), IgG3

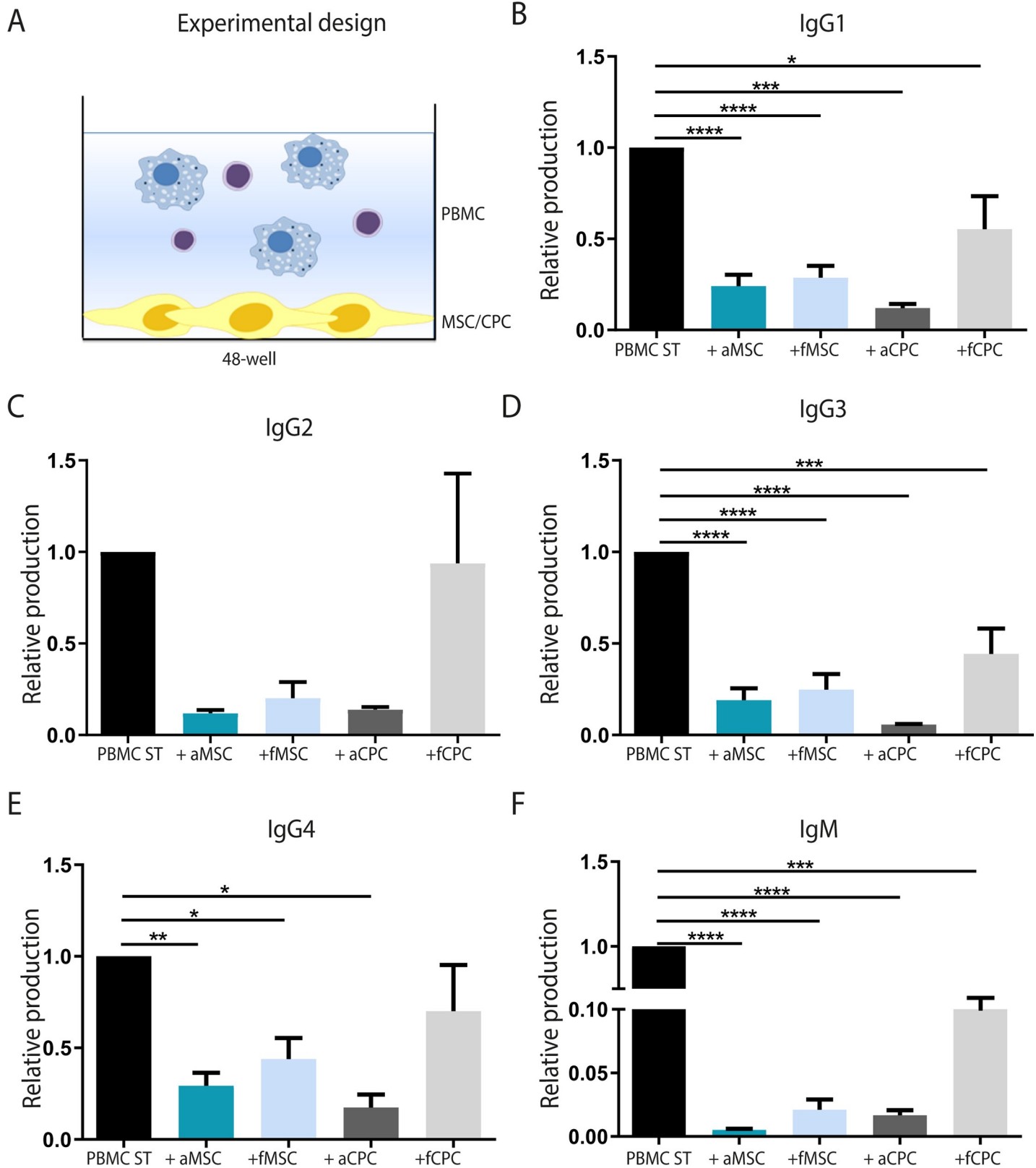

**Fig 2. Progenitor cells suppress the production of immunoglobulins.** Antibody production was measured after 10 days of co-culture with fetal or adult-derived MSCs or CPCs. **A)** Experimental design of the co-culture. **B)** Both fetal and adult MSCs showed strong immunosuppressive effects on the production of different

immunoglobulin isotypes and subclasses. IgM, IgG1, IgG3, and IgG4 levels were significantly decreased upon co-culture with MSCs. For CPCs, strongest effects were observed in cultures using adult-derived CPCs, where the production of IgM, IgG1, IgG3, and IgG4 was significantly suppressed. *aMSC: adult-derived mesenchymal stromal cell, fMSC: fetal-derived mesenchymal stromal cell, aCPC: adult-derived cardiac progenitor cell, fCPC: fetal-derived cardiac progenitor cell. For aMSC, fMSC and fCPC n = 7, for aCPC n = 3. Significance was determined using One-way ANOVA, * p<0.05, ** p<0.01, *** p<0.001, **** p<0.0001.*

(aMSC = $0.19\pm0.06$, fMSC = $0.25\pm0.09$; $p<0.0001$) and IgG4 (aMSC = $0.29\pm0.07$; $p<0.01$, fMSC = $0.43\pm0.1$; $p<0.05$). In addition, also CPCs showed strong suppressive effects on the production of mainly IgM (aCPC = $0.02\pm0.0$; $p<0.0001$, fCPC = $0.38\pm0.16$; $p<0.001$), IgG1 (aCPC = $0.12\pm0.02$; $p<0.001$, fCPC = $0.55\pm0.18$; $p<0.05$) and IgG3 (aCPC = $0.06\pm0.0$; $p<0.0001$, fCPC = $0.44\pm0.14$; $p<0.001$). For CPCs, the strongest immunosuppression was observed using adult CPCs.

## CPC-derived extracellular vesicles suppress antibody production, but are not as effective as direct cell-cell interaction when using CPCs

To explore whether the suppressive capacity of CPC on antibody production is mediated by paracrine factors, we assessed the potential of CPC-derived EVs (Fig 3). We experienced that it is technically challenging to obtain sufficient MSC-derived EVs using SEC. Therefore, we only included CPC-derived EVs in our co-cultures. Prior to co-culture, EVs were characterized based on size distribution and the presence or absence of protein markers [37]. Isolated EVs showed a representative size distribution profile with the highest peak at approximately 90 nm (Fig 3A). In line with previous findings [36], WB analysis showed that CPC-derived EVs were enriched for the typical EV proteins Alix, CD81, and CD63. Calnexin was only detectable in the cell lysate, thereby confirming the absence of contaminations with other membrane compartments (Fig 3B). An amount of 1x10 µg or 3x10 µg (every 3 days of co-culture) was added to the PBMC cultures (Fig 3C). After 10 days of co-culture, antibody secretion was significantly suppressed by EVs (Fig 3D). The production of IgM, IgG1, and IgG4 was significantly decreased using the 3x dose of CPC-derived EVs (IgM = $0.35\pm0.05$; $p<0.05$, IgG1 = $0.57\pm0.03$; $p<0.05$, and IgG4 = $0.66\pm0.0$; $p = 0.03$), thereby indicating that long term suppression is more effective than a single dose of EVs. However, the inhibitory effect was most robust when adding CPCs and not CPC-derived EVs, with strongest suppressive effects on the release of IgG1 ($0.59\pm0.1$; $p<0.05$), IgG2 ($0.23\pm0.06$; $p = 0.02$), IgG4 ($0.53\pm0.03$; $p = 0.01$) and IgM ($0.17\pm0.03$; $p<0.01$).

## MSCs show the strongest immunosuppressive effects and are more likely to be used as cell therapy in end-stage HF patients

Since we observed that CPC-derived EVs do not give the same degree of immunosuppression as CPCs, we decided to continue with CPCs to examine their potential suppressive effect on antibody-mediated immune responses in end-stage HF patients. PBMCs were isolated from end-stage HF patients and cultured with or without MSCs/CPCs. (Fig 4). At baseline culture, non-stimulated PBMCs derived from end-stage HF patients produced similar amounts of IgGs with the exception of IgG3, which is, slightly but not significantly, increased compared to PBMCs derived from healthy controls (Fig 4A). Upon co-culture of patient-derived PBMCs with MSCs or CPCs, antibody production was significantly suppressed (Fig 4B and 4C). Mainly MSCs showed strong suppressive effects, as they significantly decreased the production of IgM ($0.02\pm0.0$; $p<0.0001$), as well as all IgG subclasses (IgG1 = $0.25\pm0.08$; $p = 0.001$, IgG2 = $0.03\pm0.02$; $p<0.0001$, IgG3 = $0.25\pm0.08$; $p = 0.009$, IgG4 = $0.19\pm0.07$; $p = 0.0006$). Co-cultures using CPCs showed similar suppressions, albeit at a lower level and the differences were only statistically significant for IgM ($0.20\pm0.06$; $p<0.0001$) and IgG2 ($0.31\pm0.14$; $p = 0.0003$).

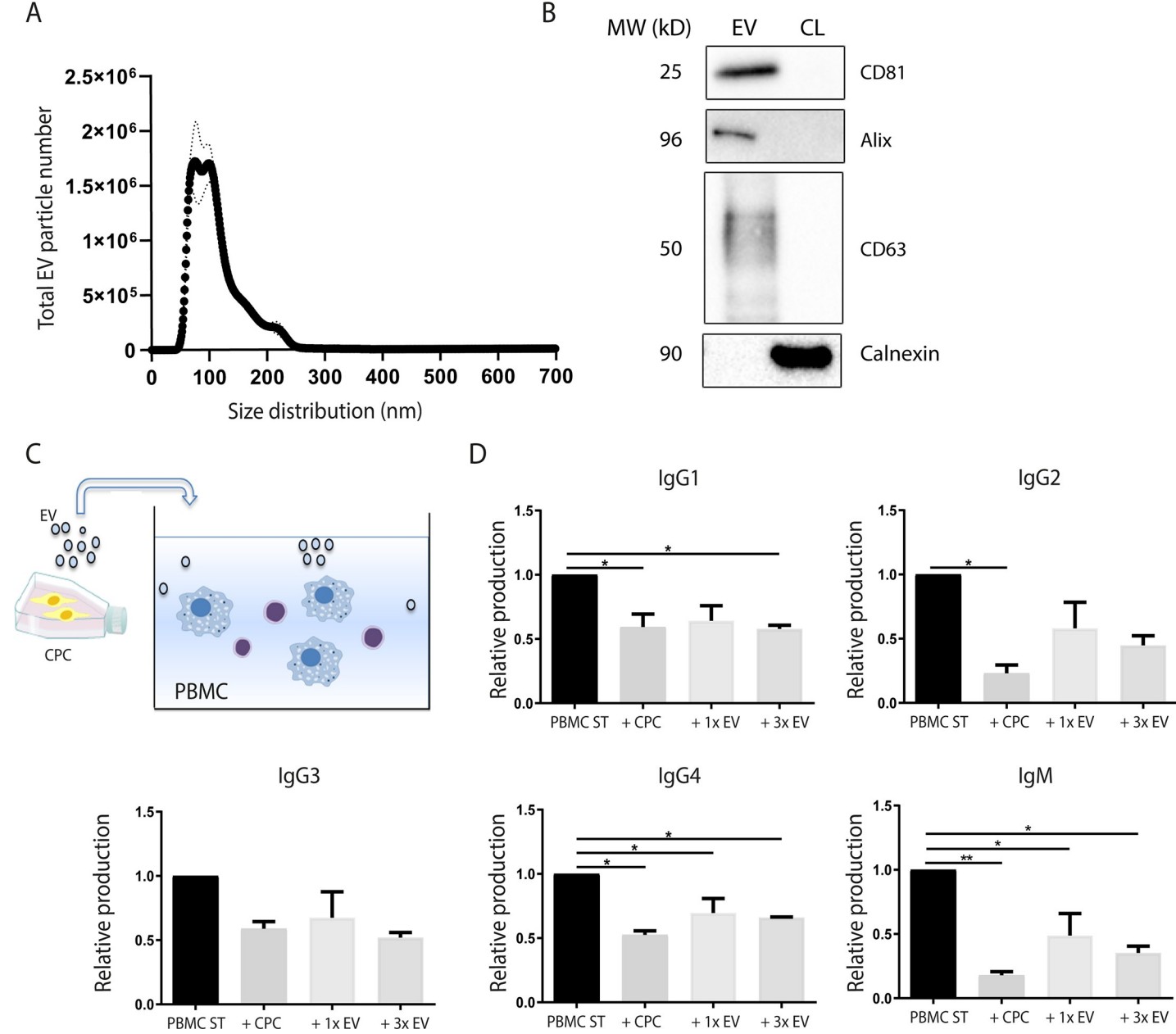

**Fig 3. Immunosuppressive capacity of CPC-derived EVs on immunoglobulin production.** To assess whether CPC-derived paracrine factors can be used, EVs were isolated and used in the PBMC co-cultures. **A)** EVs with a size of approximately 90 nm were isolated using SEC. **B)** WB of EVs and CL with general EV markers and calnexin. **C)** Experimental setup of the co-culture model, where either CPCs or CPC-derived EVs with a total of 1x 10 μg or 3x 10 μg (every 3 days of co-culture) was added to PBMCs. **D)** CPC-derived EVs showed a significant decrease of immunoglobulin production, especially when administered for a longer period of time. Levels of IgM, IgG1, and IgG4 were significantly decreased when, every 3 days of co-culture, 10 μg of EVs were added to stimulated PBMCs. However, the strongest inhibition of antibody production was observed when CPCs were used. *CPC: cardiac progenitor cell, EV: extracellular vesicles, CL: cell lysate, SEC: size-exclusion chromatography, PBMC: pheripheral blood mononuclear cells. For each condition n = 2, Significance was determined using One-way ANOVA, * p<0.05, ** p<0.01.*

## Discussion

The post-MI immune response is an important contributor to adverse cardiac remodeling and the development of HF [41–44]. The release of cardiac proteins upon MI can trigger antibody-mediated immune responses, which further induce cardiac damage and heart failure [45–47].

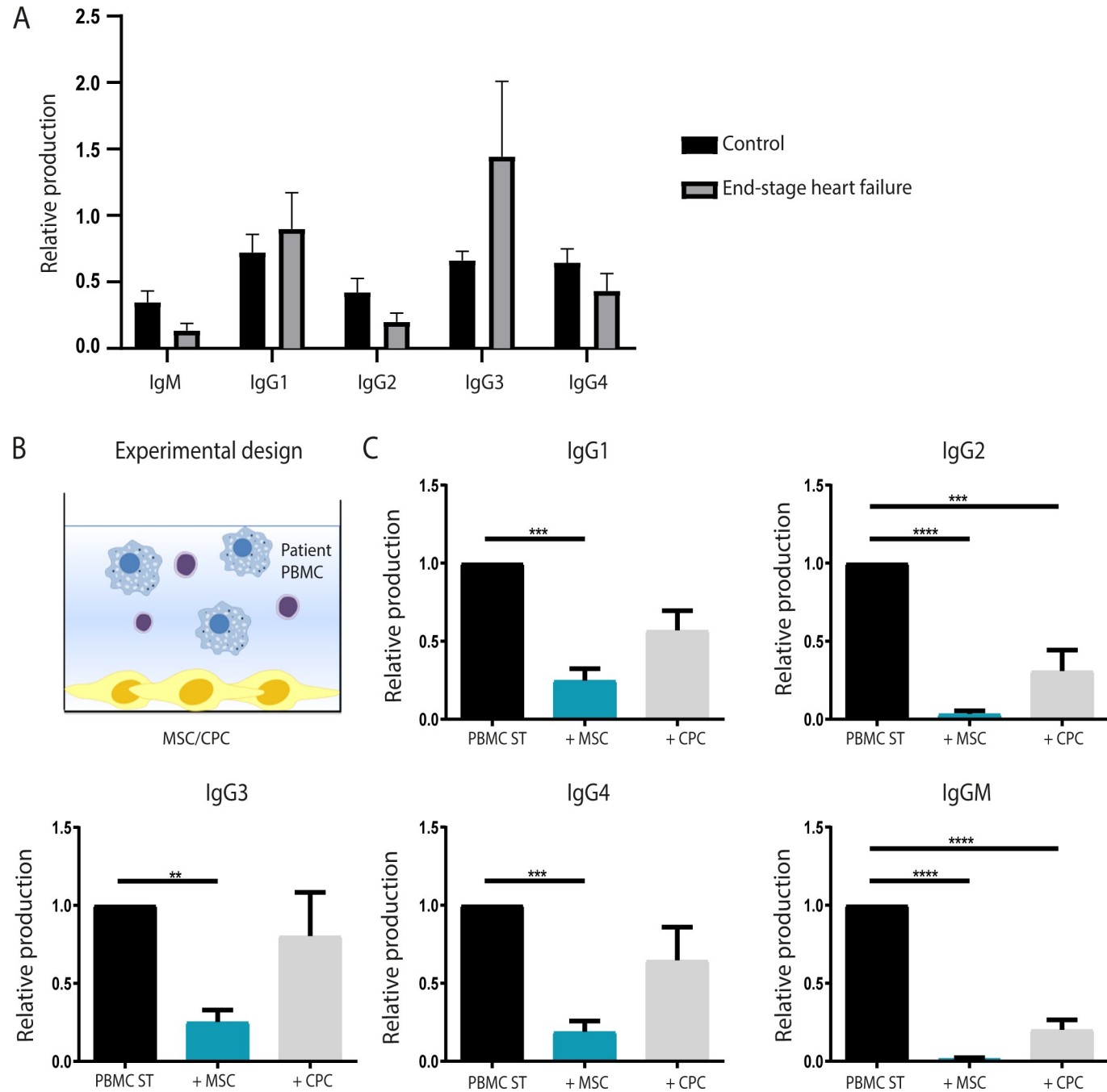

**Fig 4. Inhibition of immunoglobulin production by progenitor cells in end-stage HF.** The immunosuppressive actions of CPCs on antibody production in end-stage HF was investigated using patient-derived PBMCs. **A)** Baseline antibody levels of unstimulated PBMCs in culture were measured and compared to end-stage HF-derived PBMCs. Before co-culture with MSCs/CPCs, HF patients showed high levels of IgG1 and IgG3 compared to healthy controls. **B)** Experimental set-up of the co-culture. **C)** Levels of IgM and IgG1-IgG4 were significantly decreased upon co-culture of patient-derived PBMCs with MSCs. CPCs were able to significantly suppress IgM and IgG2, however, were not as potent as MSCs. *MSC: mesenchymal stromal cell, CPC: cardiac progenitor cell, HF: heart failure, PBMC: pheripheral blood mononuclear cells. Per condition n = 8. Significance was determined using a Kruskal Wallis test or One-way ANOVA, ** p<0.01, *** p<0.001, **** p<0.0001.*

Stem cell therapy using progenitor cells, such as MSCs or CPCs, showed promising reparative effects on cardiac function despite poor engraftment in the myocardium [17,48]. This indicates that paracrine mediators, secreted by progenitor cells, can be of great importance. MSCs and

MSC-derived EVs also have immunosuppressive properties, for example by lowering antibody production *in vitro* [49,50]. However, the immunosuppressive capacity of endogenous CPCs and CPC-derived EVs on B cells and antibody production has not been elucidated yet. Consequently in this study, we investigated the immunosuppressive effects of CPCs and CPC-derived paracrine mediators on antibody production using immune cells of both healthy controls and end-stage HF patients.

In line with previous findings, we showed that both MSCs and CPCs significantly suppressed proliferation of lymphocytes [30,32,38]. The suppressive effects of MSCs were more effective than CPCs. The suppressive effects of MSCs and CPCs on effector and regulatory T cells have been described before, where several studies show T cell inhibition via PDL-1/PD1 in a direct cell communication manner [32,51]. Moreover, both MSCs and CPCs are also able to suppress CD4+ T helper cell-mediated immune responses[52]. However, the interaction of progenitor cells with B cells is still controversial and this issue has recently gained more interest[53–55]. MSCs can inhibit plasma cell formation and subsequent IgG production in a cell-cell contact dependent as well as in an independent manner [38,55]. It is not known whether CPCs are also able to suppress antibody production *in vitro*. We demonstrated that, similar to MSCs, CPCs effectively suppress antibody production *in vitro*. We showed that both adult- and fetal-derived CPCs significantly inhibit the levels of IgM, IgG1, and IgG3, of which IgM was most efficiently suppressed, despite variation between different donors. These findings are in line with the effects of MSC, where MSC are known to exert an inhibitory effect on T helper cells, B-cell differentiation and class switching into IgG-producing cells [56,57]. Therefore, we could speculate that CPC might use a similar mechanism, in which IgG production might be suppressed either by inhibiting T-helper cell responses, thereby influencing B-cell activation and antibody production, or by directly influencing B-cell differentiation and subsequent class-switching.

To facilitate clinical translation, we examined if the strong immunosuppressive effects of CPCs and MSCs on antibody production using healthy donors, can be confirmed for IgG production using HF patient-derived PBMCs. MSCs were able to significantly inhibit the production of IgM and all IgG subclasses. For CPCs, the immunosuppressive effects were not as potent compared to MSCs, where CPCs only significantly lowered the production of IgM and IgG2. In end-stage HF, chronically activated immune cells progressively worsen cardiac function, for example by the production of cardiac antibodies [58,59]. Our findings indicate that progenitor cells, preferably MSCs, might be used as therapeutic agents to suppress antibody-mediated immune responses as observed in end-stage HF. However, mimicking the physiological immune response *in vitro*, as observed in end-stage HF patients, is still complicated. Therefore, these findings still have to be validated *in vivo*.

Part of the immunosuppressive properties of MSCs is mediated by paracrine factors, such as EVs [19,21]. The advantage of using EVs is that they can be used as a cell-free approach, thereby increasing safety, and allowing a longer duration of the treatment [19,26]. However, high variability in quantity and quality in the scaling and production process of MSC-derived EVs has been a limitation [26]. CPC-derived EVs might provide a promising alternative, not only due to their regenerative and immune modulating capacities [60], but also for their culture scalability. CPC-derived EVs have immunosuppressive effects on T cells [30,60], however, the effects on B cells and antibody-mediated responses is not clear. Our findings showed that CPC-derived EVs lower the different immunoglobulin isotypes and subclasses, such as IgM, IgG1, and IgG4. However, the number of EVs needed to reach similar suppressive effects compared to CPCs, remains challenging. In this study, we were only able to test EVs produced by fetal CPCs due to technical difficulties in obtaining sufficient numbers of EVs from adult CPCs. Fetal-derived progenitor cells might exert different effects than adult-derived cells, where, for example, adult-derived MSCs show stronger immunosuppressive capacities relative

to fetal-derived MSCs [61]. For CPCs, it has been described that fetal- and adult-derived CPCs have different developmental potentials, and adult CPCs may be more effective in cardiac repair [62,63]. In addition, fetal-derived CPCs are highly proliferative as compared to adult-derived cells. Due to this proliferative state CPCs may secrete a different palette of paracrine factors that are more associated to cell cycle rather than immunomodulation. Therefore, the effects of EVs from adult CPCs may differ from fetal–derived CPCs and have to be investigated in future studies. Nonetheless, from our data, it is clear that EVs can be used as immunosuppressive mediators, but do completely cover the strong immunosuppressive effect of CPCs.

In conclusion, we demonstrated immunosuppressive actions of both MSCs and CPCs on lymphocyte proliferation and antibody production, with strongest effects observed when using MSCs. These are partly mediated by EVs, in a time-dependent matter. Lastly, we showed that CPCs and especially MSCs were able to suppress antibody production by patient-derived cells, thereby indicating the therapeutic potential of progenitor cells in HF. Currently, cell therapy using MSCs is no longer the holy grail for true cardiac regeneration and cell replacement therapy, however, MSCs might be promising candidates for targeting the post-MI immune response and HF progression. Future studies should focus on the identification of the cardiac antigens which are targeted by the produced IgGs and on the potential of combination therapies, using both MSCs and CPCs, to simultaneously target cardiac regeneration and antibody-mediated immune responses.

## Supporting information

**S1 Fig. Extracellular vesicle markers.** Proper isolation of extracellular vesicles (EV) was determined by the presence of CD63, CD81 and Alix and absence of the cellular marker Calnexin by Western Blotting. Cell lysates (Cl) were used as controls.
(TIF)

## Acknowledgments

The authors gratefully acknowledge Erica Siera-de Koning, Joyce van Kuik, Frederieke van den Akker, and Sander van de Weg for their excellent technical support.

## Author Contributions

**Conceptualization:** Patricia van den Hoogen, Saskia C. A. de Jager, Arjan S. Schoneveld, Manon M. H. Huibers, Aryan Vink, Joost P. G. Sluijter.

**Data curation:** Patricia van den Hoogen, Emma A. Mol.

**Formal analysis:** Patricia van den Hoogen, Emma A. Mol.

**Funding acquisition:** Pieter A. Doevendans, Joost P. G. Sluijter.

**Investigation:** Patricia van den Hoogen.

**Methodology:** Patricia van den Hoogen, Saskia C. A. de Jager, Emma A. Mol, Arjan S. Schoneveld, Manon M. H. Huibers, Aryan Vink.

**Project administration:** Patricia van den Hoogen, Saskia C. A. de Jager, Joost P. G. Sluijter.

**Resources:** Manon M. H. Huibers, Aryan Vink, Joost P. G. Sluijter.

**Supervision:** Saskia C. A. de Jager, Pieter A. Doevendans, Jon D. Laman, Joost P. G. Sluijter.

**Validation:** Patricia van den Hoogen.

**Visualization:** Patricia van den Hoogen, Saskia C. A. de Jager, Emma A. Mol.

**Writing – original draft:** Patricia van den Hoogen, Joost P. G. Sluijter.

**Writing – review & editing:** Patricia van den Hoogen, Saskia C. A. de Jager, Emma A. Mol, Arjan S. Schoneveld, Manon M. H. Huibers, Aryan Vink, Pieter A. Doevendans, Jon D. Laman, Joost P. G. Sluijter.

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
