## [Decision Letter · Decision Letter 0]

31 Oct 2019

PONE-D-19-25026

Potential of mesenchymal- and cardiac progenitor cells for therapeutic targeting of B-cells and antibody responses in end-stage heart failure

PLOS ONE

Dear Ms van den Hoogen,

Thank you for submitting your manuscript to PLOS ONE. After careful consideration, we feel that it has merit but does not fully meet PLOS ONE’s publication criteria as it currently stands. Therefore, we invite you to submit a revised version of the manuscript that addresses the points raised during the review process.

We would appreciate receiving your revised manuscript by Dec 15 2019 11:59PM. To enhance the reproducibility of your results, we recommend that if applicable you deposit your laboratory protocols in protocols.io, where a protocol can be assigned its own identifier (DOI) such that it can be cited independently in the future. For instructions see: http://journals.plos.org/plosone/s/submission-guidelines#loc-laboratory-protocols

We look forward to receiving your revised manuscript.

Kind regards,

Federico Quaini, MD

Academic Editor

PLOS ONE

Journal Requirements:

3. Please include your Ethics statement in the manuscript and provide the full name of the IRB that approved the study.

Additional Editor Comments:

The study is of high interest for the scientific community because embraces different features of cardiac repair including immune mediated mechanisms. Data, however, need to be implemented to support results and conclusion. When replying to each reviewer criticism, Authors should pay attention to the major issues raised involving methodological and conceptual aspects of the work.

In addition, most of the suggested implications of your finding are related to tissue repair or cell engraftment. However, in terms of pathogenetic mechanism of heart failure, could you speculate on which target antigen these Ig are produced? For example, do you have any data on IgG or IgM against alfa- myosin heavy chain?

Although the aim of the study was to investigate humoral immunity and/or antibody mediated immune response, the possibility that changes in Th (CD4 helper) compartment could affect antibody production should be acknowledged.

The point made by reviewer #2 on patient-specific properties of CPC and MSC is of high relevance.

In line with a point made by reviewer #1, the finding that adult MSC display more immunosuppressive effect than fetal cells is understandable, while the observed similar difference in CPC is more difficult to interpret. Could you please comment on this finding? Also the observation that fetal CPC preparations are more reach in EV than the adult one might be a well known outcome although requires a comment.

Reviewers' comments:

Reviewer's Responses to Questions

**Comments to the Author**

1. Is the manuscript technically sound, and do the data support the conclusions?

Reviewer #1: Partly

Reviewer #2: Partly

2. Has the statistical analysis been performed appropriately and rigorously? 

Reviewer #1: I Don't Know

Reviewer #2: Yes

3. Have the authors made all data underlying the findings in their manuscript fully available?

Reviewer #1: No

Reviewer #2: Yes

4. Is the manuscript presented in an intelligible fashion and written in standard English?

Reviewer #1: Yes

Reviewer #2: Yes

5. Review Comments to the Author

Reviewer #1: No: PONE-D-19-25026

Potential of mesenchymal- and cardiac progenitor cells for therapeutic targeting of B-cells and antibody responses in end-stage heart failure

The manuscript is well written, the subject is relevant, the “story” well composed, and out-put a contribution to the basic understanding of MSC and CPC MoA and a next step for others to build on.

Data however need more detail to allow validation of results and conclusion.

I suggest the paper is published after revision, - provided the currently missing data upon prober presentation in fact support the claimed results.

Specific comments/questions:

Methods:

Line 113-123: Is co-culture running for 10 days with-out media change?

Line 125-138: Timing of measures is a bit blurry…. I assume CFSE labelling is performed before co-culture…

CD3 and CD19 labelling is described, - what happened to these measures, - no further evaluation is found.

Flow cytometry should be described in more detail; no gating strategy is presented

Results:

An over-all concern in the results section: I have no idea how many replicates data are based on….- and data is completely missing…. only significance levels are presented…

Which absolute/relative values were obtained?

What is the sensitivity and ranges of the IgG/M assay used, - are data in fact with-in these ranges?

Discussion:

A few more words on the difference between IgG and IgM; IgM is most efficiently suppressed. How do you interpret this finding? Any suggestions on why, and what the consequence of this finding may be?

Adult CPC are more efficient that fetal; - can this be explained?

Reviewer #2: This is a very interesting study with novel findings. However, the study has the major limitation of showing only in vitro/ex vivo data. While it is clear that requesting an in vivo study would be totally off the mark, it is however essential that at least the in vitro findings are technically undisputable to back the conclusions made.

On this premise, it would be essential that the comparison among the two progenitor cell populations was made using cells derived from the same human donor (same HLA type). From the methods section this doesn't seem to be the case for the presented experiments.

Mixing the two cell types (or their derivatives) enhances the effect of the single cell type alone? This is important because it would indirectly hypothesize whether the mechanism of action on B-cell target by the MSCs and CPCs is similar or unique to each of them.

6. PLOS authors have the option to publish the peer review history of their article (what does this mean?). If published, this will include your full peer review and any attached files.

Reviewer #1: No

Reviewer #2: No

---

## [Author Response · Author response to Decision Letter 0]

13 Dec 2019

First of all, we would like to thank the editor for the opportunity to improve our manuscript. We feel that the raised points are of additional value for our work. We have thoroughly addressed all the raised points and our response is listed below. 

Editor's Comments to Author:

1) The study is of high interest for the scientific community because embraces different features of cardiac repair including immune mediated mechanisms. Data, however, need to be implemented to support results and conclusion. When replying to each reviewer criticism, Authors should pay attention to the major issues raised involving methodological and conceptual aspects of the work.

Response: We thank the editor for the constructive remarks. We replied to each reviewer’s criticism below.

2) In addition, most of the suggested implications of your finding are related to tissue repair or cell engraftment. However, in terms of pathogenic mechanism of heart failure, could you speculate on which target antigen these Ig are produced? For example, do you have any data on IgG or IgM against alfa- myosin heavy chain?

Response: We thank the editor for this remark. We are currently working to identify the target antigens to which the IgGs are produced, which will be part of a separate manuscript. Therefore, we are currently performing a discovery epitope screening, including 26,000 antigens known in cardiovascular disease. This is still preliminary data, but depending on the different etiologies of heart failure, we identified that structural components of cardiomyocytes and cell-adhesion proteins might be affected. One of the epitopes is indeed alfa-myosin heavy chain.

We are still validating these findings in larger numbers of patients, and at this point we could only speculate that upon cardiac damage, intracellular proteins are released and exposed to the immune system, which leads to an antibody-mediated immune response against different cardiac-specific proteins, which might induce additional cardiac damage and accelerates heart failure progression. We also added an additional sentence in the coclusion:

Future studies should focus on the identification of the cardiac antigens which are targeted by the produced IgGs and on the potential of combination therapies, using both MSCs and CPCs, to simultaneously target cardiac regeneration and antibody-mediated immune responses.

3) Although the aim of the study was to investigate humoral immunity and/or antibody mediated immune response, the possibility that changes in Th (CD4 helper) compartment could affect antibody production should be acknowledged.

Response: We agree that differences in CD4 Th responses between donors may have directly altered antibody production. 

It has been established that MSCs (and CPCs) can also affect CD4 T cell responses and as a consequence B cell mediated antibody production. We cannot exclude that antibody production in our experiment was indirectly altered by differences in CD4 T cell compartment. Although we did not assess CD4 T cell numbers after 10 days of co-culture, we did assess CD4-T cell numbers at baseline. We did not observe differences in the number of CD4 T cells between donors at baseline, which may suggest the CD4 T compartment will also remain similar upon stimulation. We have now touched upon this in the discussion (line 250-268):

In line with previous findings, we showed that both MSCs and CPCs significantly suppressed proliferation of lymphocytes (30,32,38). The suppressive effects of MSCs were more effective than CPCs. The suppressive effects of MSCs and CPCs on effector and regulatory T cells have been described before, where several studies show T cell inhibition via PDL-1/PD1 in a direct cell communication manner (32,51). Moreover, both MSCs and CPCs are also able to suppress CD4+ T helper cell-mediated immune responses(52). However, the interaction of progenitor cells with B cells is still controversial and this issue has recently gained more interest(53–55). MSCs can inhibit plasma cell formation and subsequent IgG production in a cell-cell contact dependent as well as in an independent manner (38,55). It is not known whether CPCs are also able to suppress antibody production in vitro. We demonstrated that, similar to MSCs, CPCs effectively suppress antibody production in vitro. We showed that both adult- and fetal-derived CPCs significantly inhibit the levels of IgM, IgG1, and IgG3, of which IgM was most efficiently suppressed, despite variation between different donors. These findings are in line with the effects of MSC, where MSC are known to exert an inhibitory effect on T helper cells, B-cell differentiation and class switching into IgG-producing plasma cells(56,57). Therefore, we could speculate that CPC might use a similar mechanism, in which IgG production might be suppressed either by inhibiting T-helper cell responses, thereby influencing B-cell activation and antibody production, or by directly influencing B-cell differentiation and subsequent class-switching.

4) The point made by reviewer #2 on patient-specific properties of CPC and MSC is of high relevance. 

Response: we agree with the editor and reviewer #2. We completely understand and agree with the comment of the reviewer, that using cells derived from the same donor would be the most optimal approach to make the best comparison between the different cell types. Unfortunately, due to ethical reasons, we are not allowed to harvest the two types of progenitor cells (MSC and CPC) from the same donor. The CPCs are harvested upon coronary artery bypass grafting (CABG), which is an invasive procedure that in itself is already challenging for both the patient and surgeon. The CPCs are directly harvested form the heart, which is technically very challenging and comes with additional risk for the patient and is only obtained from planned operations and considered waist material. 

5) In line with a point made by reviewer #1, the finding that adult MSC display more immunosuppressive effect than fetal cells is understandable, while the observed similar difference in CPC is more difficult to interpret. Could you please comment on this finding? 

Response: In contrast with fetal CPC, which are obviously isolated from fetal tissue, adult CPC are harvested from patients with ischemic heart disease that are eligible for coronary artery bypass grafting (CABG). The CPCs are operatively collected during the CABG procedure. We could therefore speculate that these cells are already primed, as a consequence of chronic inflammation associated to atherosclerosis or oxygen deprivation due to ischemia. The priming will result in activated state that may lead to increased secretion of paracrine factors, which in this case, could result in a stronger immunosuppressive capacity as compared to fetal-derived CPC. Moreover, fetal-derived cells are highly proliferative as compared to adult-derived cells. Due to this proliferative state CPCs may secrete a different palette of paracrine factors that are more associated to cell cycle rather than immunomodulation. This is still subject for further study. We have now touched upon this in the discussion (line 295-298):

For CPCs, it has been described that fetal- and adult-derived CPCs have different developmental potentials, and adult CPCs may be more effective in cardiac repair (62,63). In addition, fetal-derived CPCs are highly proliferative as compared to adult-derived cells. Due to this proliferative state CPCs may secrete a different palette of paracrine factors that are more associated to cell cycle rather than immunomodulation. Therefore, the effects of EVs from adult CPCs may differ from fetal–derived CPCs and have to be investigated in future studies.

6) Also the observation that fetal CPC preparations are more reach in EV than the adult one might be a well-known outcome although requires a comment.

Response: We thank the Editor for this remark. We believe this may be directly related to the point raised above. Fetal-derived CPC are highly proliferative and grow must faster than adult-derived CPC. This could also influence the production of extracellular vesicles and the secretion of paracrine factors. We observed higher yields of EV production by fetal-derived cells, which may suggest that fetal-derived CPC can cope with medium starvation better compared to adult CPC.

 

Reviewer(s)' Comments to Author:

Reviewer: 1

We thank reviewer 1 for his/her constructive remarks. Answers to the questions are displayed in Italics. All changes and additions to the manuscript are underlined.

Comments to the Author 

The manuscript is well written, the subject is relevant, the “story” well composed, and out-put a contribution to the basic understanding of MSC and CPC MoA and a next step for others to build on. Data however need more detail to allow validation of results and conclusion. I suggest the paper is published after revision, - provided the currently missing data upon prober presentation in fact support the claimed results.

Methods: 

1) Line 113-123: Is co-culture running for 10 days with-out media change?

Response: Indeed, the co-culture is running for 10 days, without media change to make sure that all immunoglobulins which are produced are in the supernatant upon collection. We choose 10 days, based on the fact that the production of immunoglobulins normally takes place after 7-10 days after initial B-cell activation. We added some additional words in our methods section:

PBMCs were co-cultured for 10 days, without medium change, in 48-well plates (2.5x105 PBMC/well) with MSCs, CPCs (5.0x104 cells/well), or CPC-derived EVs (1x 10 µg= 6.3x1010 particles), immediately upon co-culture or 3x 10 µg (6.3x1010 particles) added every 3 days of co-culture.

2) Line 125-138: Timing of measures is a bit blurry…. I assume CFSE labelling is performed before co-culture…

Response: The reviewer is right, the CFSE labeling is performed after PBMC isolation and before the co-culture. We added this to our method section:

Flow cytometry (Gallios, Beckmann Coulter) was used to assess lymphocyte proliferation. Prior to co-culture, PBMCs were labeled with 1.5mM carboxyfluorescein succinimidyl ester (CFSE, Sigma, 21888) as described previously (33)

3) CD3 and CD19 labelling is described, - what happened to these measures, - no further evaluation is found. Flow cytometry should be described in more detail; no gating strategy is presented

Response: We understand the reviewer’s remark, indeed the data about CD3+ and CD19+ lymphocytes is not incorporated in our manuscript. After isolation and prior to co-culture, we stained the cells for CD3 and CD19 in order to assess the relative contribution of T- and B- cells to the lymphocyte population of each donor. We assessed whether the cell populations are within normal range and similar between the different donors, just to make sure we started with the same number B cells at the start of the co-culture and the differences in immunoglobulins after 10 days cannot be explained by differences in the number of B cells prior to co-culture. We did not specifically stain for CD3 and CD19 at termination of the co-culture. We added an additional sentence in our method sections:

After two washing steps with PBS, PBMCs were incubated for 30 min with fluorescent antibodies, including CD3 for T cells (Brilliant Violet 510, Biolegend, 317332) and CD19 for B cells (PE/Cy5, Biolegend 302210). After washing with PBS, PBMCs were incubated for 30 min with a fixable viability dye (eFluor506, Bioscience, 65-0866-14) to exclude dead cells. Prior to culture, general cell composition per donor was assessed by measuring the percentage of CD3+ T cells and CD19+ B cells , to ensure that the cell populations were similar between the different donors at baseline. Lymphocyte proliferation was calculated by measuring CFSE intensity and the number of cells present in each division as described before (33).

Moreover, we added an informative figure below, including the gating strategy to identify the number of T- and B cells, and PBMC cell composition and the percentage of T and B cells at baseline prior to co-culture. If the reviewer feels that this figure is of great value for our manuscript, we can include this as a supplementary figure in our manuscript.

Results

4) An over-all concern in the results section: I have no idea how many replicates data are based on….- and data is completely missing…. only significance levels are presented…

Response: With respect, we don’t understand which data the reviewer refers to when stating data is completely missing. All the data related to this work is part of the manuscript or supplemental data. We do apologize that the number of replicates was not described clearly enough. In the original paper we touch upon that in the figure legends, were we stated the number of PBMC donors per experiment. Each individual PBMC donor is considered as independent experimental number (n), ranging from 2-8 donors per experiment. We now also added this in our statistics section: 

Statistical analysis and data representation were performed using IBM SPSS Statistics 21 and Graphpad Prism© (GraphPad Software Inc. version 8.01, San Diego CA, USA). Normal data distribution was tested using the Kolmogorov-Smirnov test. Group comparison was performed by a one-way ANOVA or Kruskal-Wallis test, corrected for multiple comparison testing. Each individual PBMC donor is considered as an independent individual experimental number (n), ranging from 2-8 donors per experiment. Data was considered significant with two-tailed p-values <0.05 and is presented as mean ± SEM.

5) Which absolute/relative values were obtained?

Response: Due to donor differences in the absolute values of immunoglobulin levels, we decided to use relative values and normalize our data per donor. PBMCs stimulated with IL-2 and PMA, without addition of MSCs, CPCs or EVs, served as our positive control and was defined as maximal stimulation (100%). This condition was used as a reference to normalize the other conditions per donor and per experiment. Meaning for each individual donor a maximum response was determined and the suppressive capacity of MSC, CPCs or EVs is depicted relative (as ratio) to this maximal response. We clarified this in our method section and in addition to the significance levels, we also added the relative values in our results section:

Production of IgM and different IgG subclasses is suppressed by cardiac-derived progenitor cells.

Next to reduced cell proliferation, MSCs are also able to inhibit several immune cell functions, such as antibody secretion (38). To examine whether this also holds true for CPCs, we collected the supernatant after 10 days of co-culture and measured the levels of different immunoglobulin subclasses (Fig 2A). Since it is known that the age of the donor can affect their inhibitory potency (39,40), both fetal and adult MSCs and CPCs were included. Adult and fetal-derived MSCs significantly inhibited antibody production from stimulated PBMCs (Fig 2B-F). Fetal and adult MSC significantly reduced the production of IgM (aMSC=0.005±0.0 fMSC=0.02±0.0; p<0.0001), IgG1 (aMSC=0.24±0.06, fMSC=0.28±0.06; p<0.0001), IgG3 (aMSC=0.19±0.06, fMSC=0.25±0.09; p<0.0001) and IgG4 (aMSC=0.29±0.07; p<0.01, fMSC=0.43±0.1; p<0.05). In addition, also CPCs showed strong suppressive effects on the production of mainly IgM (aCPC=0.02±0.0; p<0.0001, fCPC=0.38±0.16; p<0.001), IgG1 (aCPC=0.12±0.02; p<0.001, fCPC=0.55±0.18; p<0.05) and IgG3 (aCPC=0.06±0.0; p<0.0001, fCPC=0.44±0.14; p<0.001). For CPCs, the strongest immunosuppression was observed using adult CPCs.

CPC-derived extracellular vesicles suppress antibody production, but are not as effective as direct cell-cell interaction when using CPCs

The production of IgM, IgG1, and IgG4 was significantly decreased using the 3x dose of CPC-derived EVs (IgM=0.35±0.05; p<0.05, IgG1=0.57±0.03; p<0.05, and IgG4=0.66±0.0; p=0.03), thereby indicating that long term suppression is more effective than a single dose of EVs. However, the inhibitory effect was most robust when adding CPCs and not CPC-derived EVs, with strongest suppressive effects on the release of IgG1 (0.59±0.1; p<0.05), IgG2 (0.23±0.06; p=0.02), IgG4 (0.53±0.03; p=0.01) and IgM (0.17±0.03; p<0.01).

MSCs show the strongest immunosuppressive effects and are more likely to be used as cell therapy in end-stage HF patients

Upon co-culture of patient-derived PBMCs with MSCs or CPCs, antibody production was significantly suppressed (Fig. 4B-C). Mainly MSCs showed strong suppressive effects, as they significantly decreased the production of IgM (0.02±0.0; p<0.0001), as well as all IgG subclasses (IgG1=0.25±0.08; p=0.001, IgG2=0.03±0.02; p<0.0001, IgG3=0.25±0.08; p=0.009, IgG4=0.19±0.07; p=0.0006). Co-cultures using CPCs showed similar suppressions, albeit at a lower level and the differences were only statistically significant for IgM (0.20±0.06; p<0.0001) and IgG2 (0.31±0.14; p=0.0003).

6) What is the sensitivity and ranges of the IgG/M assay used, - are data in fact with-in these ranges?

Response: We made use of the Bio-plex Pro Human Isotyping assay. In this assay, a quality control is included to ensure whether the assay was performed correctly and whether the measured values were in the normal ranges according to the assay. Proper sample dilution was first tested in each donor, ensuring all samples were in the linear part of the standard curve and thus within the detection range of the assay. 

We added the assay performance according to the manufacturer below. We can confirm that our data was in fact within these ranges, when samples were diluted 5 (we also added this in the method section).

The levels of IgM and IgG subclasses (IgG1, IgG2, IgG3, IgG4) in the co-cultures (5x diluted) were measured using a Bio-Plex Pro™ human isotyping immunoassay 6-plex (Bio-Rad, 171A3100M) according to manufacturer’s instructions and were all within the detection limit of the assay.

Biorad: https://www.bio-rad.com/webroot/web/pdf/lsr/literature/Bulletin_6344.pdf

Discussion

7) A few more words on the difference between IgG and IgM; IgM is most efficiently suppressed. How do you interpret this finding? Any suggestions on why, and what the consequence of this finding may be?

Response: We thank the reviewer for this remark. With the approach we now used, we mainly aimed to assess if antibody production by B cells could be altered by MSC or CPCs we therefore chose to stimulate PBMC with PMA, which is a general activator of lymphocytes and also stimulates B-cell division and antibody production. IgM is the first antibody produced in great amounts upon activation, which in vivo is followed by the production of antigen-specific IgG after class-switching. Since we use a non-antigen specific stimulus (PMA) in our opinion it is not un-logical that we observe the most robust effect on IgM production. We expect the effect on IgGs may have been more pronounced when stimulating the cells with cardiac derived antigens, which a). are likely different between donors and b) have not been fully elucidated yet. In patients with heart failure we observe that mostly IgG is increased in the circulation and in the heart of end-stage heart failure. Obviously, more work is needed to determine patient specific epitope signatures (beyond the scope of this manuscript), but IgG in general and more specifically antigen specific IgGs may be considered as biomarkers for disease progression or even targets for therapeutic interventions in end-stage HF.

Indeed, the levels of IgM appear to be more efficiently suppressed in this in vitro setting compared to IgG. We did not look into the exact effects of progenitor cells on B-cell differentiation and class switching in this manuscript, and we could therefore only speculate that these findings might indicate that our progenitor cells might already have suppressive effects on B-cell differentiation and class switching, which in the end also results in lower levels of IgG. We added some additional sentences in the discussion of our manuscript:

We demonstrated that, similar to MSCs, CPCs effectively suppress antibody production in vitro. We showed that both adult- and fetal-derived CPCs significantly inhibit the levels of IgM, IgG1, and IgG3, of which IgM was most efficiently suppressed, despite variation between different donors. These findings are in line with the effects of MSC, where MSC are known to exert an inhibitory effect on B-cell differentiation and class switching into IgG-producing cells (56,57). Therefore, we could speculate that CPC might use a similar mechanism, in which IgG production might be suppressed either by inhibiting T-helper cell responses, thereby influencing B-cell activation and antibody production, or by directly influencing B-cell differentiation and subsequent class-switching.

8) Adult CPC are more efficient that fetal; - can this be explained?

Response: In contrast with fetal CPC, which are obviously isolated from fetal tissue, adult CPC are harvested from patients with ischemic heart disease that are eligible for coronary artery bypass grafting (CABG). The CPCs are operatively collected during the CABG procedure. We could therefore speculate that these cells are already primed, as a consequence of chronic inflammation associated to atherosclerosis or oxygen deprivation due to ischemia. The priming will result in activated state that may lead to increased secretion of paracrine factors, which in this case, could result in a stronger immunosuppressive capacity as compared to fetal-derived CPC. Moreover, fetal-derived cells are highly proliferative as compared to adult-derived cells. Due to this proliferative state CPCs may secrete a different palette of paracrine factors that are more associated to cell cycle rather than immunomodulation. This is still subject for further study. We have now touched upon this in the discussion (line 295-298):

For CPCs, it has been described that fetal- and adult-derived CPCs have different developmental potentials, and adult CPCs may be more effective in cardiac repair (62,63). In addition, fetal-derived CPCs are highly proliferative as compared to adult-derived cells. Due to this proliferative state CPCs may secrete a different palette of paracrine factors that are more associated to cell cycle rather than immunomodulation. Therefore, the effects of EVs from adult CPCs may differ from fetal–derived CPCs and have to be investigated in future studies.

Reviewer: 2

We thank the reviewer for his/her constructive remarks. Answers to the questions are displayed in Italics. All changes and additions to the manuscript are underlined.

1) This is a very interesting study with novel findings. However, the study has the major limitation of showing only in vitro/ex vivo data. While it is clear that requesting an in vivo study would be totally off the mark, it is however essential that at least the in vitro findings are technically undisputable to back the conclusions made.

Response: We agree with the reviewer. Validating our findings in an in vivo model is also in our opinion the next step and we are currently working on the details and future experiments.

2) On this premise, it would be essential that the comparison among the two progenitor cell populations was made using cells derived from the same human donor (same HLA type). From the methods section this doesn't seem to be the case for the presented experiments.

Response: we agree with the editor and reviewer #2. We completely understand and agree with the comment of the reviewer, that using cells derived from the same donor would be the most optimal approach to make the best comparison between the different cell types. Unfortunately, due to ethical reasons, we are not allowed to harvest the two types of progenitor cells (MSC and CPC) from the same donor. The CPCs are harvested upon coronary artery bypass grafting (CABG), which is an invasive procedure that in itself is already challenging for both the patient and surgeon. The CPCs are directly harvested form the heart, which is technically very challenging and comes with additional risk for the patient and is only obtained from planned operations and considered waist material. 

3) Mixing the two cell types (or their derivatives) enhances the effect of the single cell type alone? This is important because it would indirectly hypothesize whether the mechanism of action on B-cell target by the MSCs and CPCs is similar or unique to each of them.

Response: We thank the reviewer for this suggestion. We agree with the reviewer, however, we encountered technical difficulties in mixing the same cell types in our in vitro co-culture. MSC and CPC both have their own specific medium and culture protocols, therefore, combining these two cell types in one well cannot be performed without creating and validating a new culture protocol, to make sure that the functional and paracrine effects of both cell types are not altered or influenced. Moreover unfortunately, it is not possible to obtain MSCs and CPCs from the same human donor. Therefore, as a consequence, combining MSCs and CPCs would result in mixing two different HLA types/ donors, which in our opinion is not favorable. 

Mixing two types of extracellular vesicles (EVs) is a nice suggestion and would be a good addition to our paper. However, we encountered technical difficulties in obtaining sufficient numbers of MSC-derived EVs. In order to purify our EV population, we cultured our MSCs in serum-free culture media before EV collection and isolation. However, MSCs cultured in serum-free medium, in our hands, fail to produce sufficient numbers of EVs for further analysis. Therefore, it is not possible to combine MSC-EV with CPC-EVs.

The use of cell-derived conditioned medium is in our opinion, the only possible way to combine the derivatives of the two cell types. However, we expect this will not be directly comparable with our EV data, since conditioned medium contains many other (un)identified soluble paracrine factors secreted by the cells next to EVs. 

Therefore, we feel the suggestion is of great additional value and was therefore also suggested as potential future therapeutic direction in our discussion, but due to the technical limitations described above, we cannot answer this question at the moment.

---

## [Editor Report · Decision Letter 1]

17 Dec 2019

Potential of mesenchymal- and cardiac progenitor cells for therapeutic targeting of B-cells and antibody responses in end-stage heart failure

PONE-D-19-25026R1

Dear Dr. van den Hoogen,

We are pleased to inform you that your manuscript has been judged scientifically suitable for publication and will be formally accepted for publication once it complies with all outstanding technical requirements.

With kind regards,

Federico Quaini, MD

Academic Editor

PLOS ONE

Additional Editor Comments (optional):

The Authors have properly reply to each criticisms and, by doing this, the scientific content of the manuscript has significantly improved.
---

## [Editor Report · Acceptance letter]

20 Dec 2019

PONE-D-19-25026R1 

Potential of mesenchymal- and cardiac progenitor cells for therapeutic targeting of B-cells and antibody responses in end-stage heart failure 

Dear Dr. van den Hoogen:

I am pleased to inform you that your manuscript has been deemed suitable for publication in PLOS ONE. Congratulations! Your manuscript is now with our production department. 

With kind regards,

on behalf of

Dr. Federico Quaini 

Academic Editor

PLOS ONE